# Multimodal Imaging Techniques to Evaluate the Anticancer Effect of Cold Atmospheric Pressure Plasma

**DOI:** 10.3390/cancers13102483

**Published:** 2021-05-19

**Authors:** Marcel Kordt, Isabell Trautmann, Christin Schlie, Tobias Lindner, Jan Stenzel, Anna Schildt, Lars Boeckmann, Sander Bekeschus, Jens Kurth, Bernd J. Krause, Brigitte Vollmar, Eberhard Grambow

**Affiliations:** 1Rudolf-Zenker-Institute of Experimental Surgery, Rostock University Medical Center, 18057 Rostock, Germany; isabell.trautmann@uni-rostock.de (I.T.); christin.schlie@med.uni-rostock.de (C.S.); brigitte.vollmar@med.uni-rostock.de (B.V.); eberhard.grambow@med.uni-rostock.de (E.G.); 2Core Facility Multimodal Small Animal Imaging, Rostock University Medical Center, 18057 Rostock, Germany; tobias.lindner@med.uni-rostock.de (T.L.); jan2.stenzel@gmail.com (J.S.); anna.schildt@med.uni-rostock.de (A.S.); bernd.krause@med.uni-rostock.de (B.J.K.); 3Clinic and Policlinic for Dermatology and Venereology, Rostock University Medical Center, 18057 Rostock, Germany; lars.boeckmann@med.uni-rostock.de; 4Center for innovation competence (ZIK) *plasmatis*, Leibniz Institute for Plasma Science and Technology (INP), 17489 Greifswald, Germany; sander.bekeschus@inp-greifswald.de; 5Department of Nuclear Medicine, Rostock University Medical Center, 18055 Rostock, Germany; Jens.Kurth@med.uni-rostock.de; 6Department for General, Visceral-, Vascular- and Transplantation Surgery, Rostock University Medical Center, 18057 Rostock, Germany

**Keywords:** kINPen™, malignant melanoma, plasma medicine, reactive oxygen and nitrogen species, skin cancer, squamous cell carcinoma

## Abstract

**Simple Summary:**

Skin cancer still poses a great burden for patients. One of the most promising therapy approaches in recent years is cold atmospheric pressure plasma. The aim of the study was to assess in vivo the effect of cold atmospheric pressure plasma on human squamous cell carcinoma as well as malignant melanoma tumour cell lines in a longitudinal and non-invasive manner with multimodal imaging techniques such as MRI and [18F]FDG PET. By using these techniques and chemiluminescence imaging, we present in the current study that reactive species increase in vivo upon treatment with cold plasma and consequently decrease tumour growth. Therefore, we propose cold atmospheric pressure plasma may be a potential adjuvant therapy option to established standard therapies of skin cancer.

**Abstract:**

Background: Skin cancer is the most frequent cancer worldwide and is divided into non-melanoma skin cancer, including basal cell carcinoma, as well as squamous cell carcinoma (SCC) and malignant melanoma (MM). Methods: This study evaluates the effects of cold atmospheric pressure plasma (CAP) on SCC and MM in vivo, employing a comprehensive approach using multimodal imaging techniques. Longitudinal MR and PET/CT imaging were performed to determine the anatomic and metabolic tumour volume over three-weeks in vivo. Additionally, the formation of reactive species after CAP treatment was assessed by non-invasive chemiluminescence imaging of L-012. Histological analysis and immunohistochemical staining for Ki-67, ApopTag^®^, F4/80, CAE, and CD31, as well as protein expression of PCNA, caspase-3 and cleaved-caspase-3, were performed to study proliferation, apoptosis, inflammation, and angiogenesis in CAP-treated tumours. Results: As the main result, multimodal in vivo imaging revealed a substantial reduction in tumour growth and an increase in reactive species after CAP treatment, in comparison to untreated tumours. In contrast, neither the markers for apoptosis, nor the metabolic activity of both tumour entities was affected by CAP. Conclusions: These findings propose CAP as a potential adjuvant therapy option to established standard therapies of skin cancer.

## 1. Introduction

Skin cancer is the most common form of cancer worldwide [1,2]. Cancer of the skin is commonly divided into non-melanoma skin cancer (NMSC) and malignant melanoma (MM). The major subtypes of NMSC include basal cell carcinoma (BCC) and squamous cell carcinoma (SCC) [2]. Estimates assume that there were nearly 1.2 million new cases of NMSC worldwide in 2020 alone, and almost 325 thousand people developed MM. In 2020, it is estimated that approximately 64 thousand and 57 thousand patients died worldwide from NMSC and MM, respectively [3]. Most deaths are caused by MM, although it accounts for only 2% of malignant skin cancers [1]. The tumour stage determines the prognosis for patients with skin cancer at the initial diagnosis and the presence of lymph node metastases and distant metastases. Nevertheless, the relative five-year survival rate for all stages together for the MM is 92%, whereas the relative five-year survival rate for MM with distant metastases drops to 25% [4]. State of the art therapy, according to the tumour stage, is often a combination of surgery, chemotherapy, and irradiation. Metastatic skin tumours are frequently treated with chemotherapy [5,6]. In 2006 alone, more than two million people in America were treated for BCC and SCC [7]. In the last decade, new therapeutic options have been developed. After the US Food and Drug Administration approved new therapies for metastatic diseases, such as the antibody against CTLA-4 ipilimumab and the BRAF inhibitor vemurafenib in 2011, there was a decline in mortality in MM. For men and women aged 20 to 64, the overall mortality rate was reduced by 7% annually between 2013 and 2017 [4]. Despite the established therapies and the newly approved ones, additional therapy options are needed to further improve survival outcomes and the quality of life of patients with malignant skin cancer.

For patients with skin cancer, various imaging techniques, such as ultrasound, computed tomography (CT), magnetic resonance imaging (MRI), or positron emission tomography (PET) have become an integral part of the diagnosis [8,9]. To ensure that patients receive the most appropriate and effective treatment, it is relevant to identify the correct tumour stage, i.e., whether the tumour has spread and to which parts of the body. The ^18^F-2-fluoro-2-deoxy-d-glucose ([^18^F]FDG) PET, especially, is an essential component for the assessment of later stages (stage III) and the metastatic spreading and monitoring of the response to therapy of NMSC and MM [8,9,10]. Patients with MM are also screened with MRI to confirm distant metastases in the brain [10,11].

One of the most promising approaches to cancer therapy in recent years, especially in skin cancer, is cold atmospheric pressure plasma (CAP). Physical plasma, the so-called fourth state of matter, usually exhibits temperatures much higher than what is tolerated by biological systems. This is due to electrons and heavy particles, such as atoms and ions, being in thermal equilibrium. This equilibrium occurs when heavy particles and electrons move and collide intensely [12]. In cold plasma, heavy particles are discharged faster than electrons, a thermal imbalance occurs and plasma reaches a temperature of 25 to 45 °C, allowing its use for application in humans [13]. The two most important CAP devices in clinical use are the dielectric barrier discharge and the plasma jet. Several studies showed the significant anticancer capacity and therapeutic potential of CAP in over dozens of in vitro cancer cell lines, and in several in vivo subcutaneous (s.c.) xenograft tumours, such as head and neck cancer, skin cancer, colorectal cancer, lung cancer, breast cancer, as well as bladder cancer [6,13,14,15]. In line with these observations, cancer therapy with CAP has already been used with great promise in the human field [16,17]. In contrast to established therapy options, such as surgery and anticancer drugs, CAP is a selective method of treatment against cancer cells [13]. This effect on cancer cells is mediated by a large intracellular increase in reactive oxygen species (ROS) and reactive nitrogen species (RNS) in tumour cells [18,19].

Despite the knowledge gained in recent years, a comprehensive view on tumour response upon CAP treatment is still missing. Therefore, this study focused on the in vivo effect of a high-frequency CAP source on s.c. SCC and MM. CAP effects were assessed longitudinally using non-invasive multimodal imaging techniques, such as MRI and PET/CT. Additionally, chemiluminescence (CL) imaging was performed to quantify ROS in s.c. tumours upon CAP treatment.

## 2. Methods

### 2.1. Cell Culture

The human SCC cell line A431 and the MM cell line A375 were used for the in vivo experiments (Cell Lines Service GmbH (CLS), Eppelheim, Germany). The cells were cultured under standard cell culture conditions at 37 °C and 5% CO_2,_ in Dulbecco’s modified Eagle’s serum GlutaMAX™ (DMEM GlutaMAX™, Gibco, Thermo Fisher Scientific, Waltham, MA, USA), supplemented with 10% foetal calf serum (FCS, Biochrom GmbH, Berlin, Germany), 100U/mL penicillin and 100 mg/mL streptomycin (Sigma-Aldrich, Taufkirchen, Germany). For in vivo experiments, cells were harvested with 1% Trypsin/EDTA and stored on ice. After quantification of cell numbers, cells were suspended on ice with 1:1 cold DPBS/Matrigel^®^ High Concentration Growth Factor Reduced (Matrigel^®^ HC GFR, Corning, New York, NY, USA). Cell lines in a passage between 35 and 40 were used for the in vivo experiments.

### 2.2. Ethics Statement

NOD.Cg-Prkdc^scid^ Il2rg^tm1Wjl^/SzJ (NSG) mice were initially purchased from Jackson Laboratory. These mice developed immunodeficiency, due to the severe combined immune deficiency and a complete null allele of the IL2 receptor common gamma chain. NSG mice were kept at a 12 h light/dark cycle and an ambient temperature of 21 ± 2 °C with 60 ± 20% relative humidity. Water and standard laboratory chow were provided ad libitum. Mice were bred under specific germ-free conditions (except for the following pathogens detected within the last two years, i.e., *Helicobacter sp.*, *Rodentibacter pneumotropicus*, *murine Norovirus* and *rat Theilovirus*) in individually ventilated cages and with environmental enrichment [20]. Mice were group-housed during breeding and for the experiments, different numbers of mice (*N* = 2–5) were housed in cages and different litters were used. Before the experiment, the mice had one week for habituation. For all experiments, male NSG mice with an age of 8 to 20 weeks and a bodyweight between 20 and 30g were used. Please note that the focus on male immunodeficient NSG mice could be a limitation of this study. The required number of mice was calculated before starting the experiments, by sample size calculation (alpha = 0.05, beta = 0.20, power = 0.8). This study and the animal experiments were approved by the State Department for Agriculture, Food Safety and Fishery in Mecklenburg–Western Pomerania (7221.3-1-057/18) and were conducted in accordance with the German law for animal protection (TierSchG) and the EU Directive 2010/63/EU. The current study is a part of the joint research project, ONKOTHER-H, and is funded by the European Social Fund (ESF, grant numbers ESF/14-BM-A55-0003/18) and the Ministry of Education, Science and Culture of Mecklenburg–Vorpommern, Germany. In this context, a project protocol was prepared by all members of the ONKOTHER-H group before the project started. However, this protocol was not officially registered.

### 2.3. Cold Atmospheric Pressure Plasma Device

The CAP device was supplied by the Leibniz Institute for Plasma Science and Technology (INP Greifswald, Greifswald, Germany). It operates at temperatures between 35 and 50 °C [21]. For the treatment of s.c. tumours, the high-frequency plasma jet kINPen™ IND (neoplas GmbH, Greifswald, Germany) was used. The jet was operated with argon as feed gas at a flow rate of 5 litres per minute. The operating distance from the pencil to the skin surface of the tumour-bearing mouse was between 0.8 and 1 cm. The kINPen IND plasma jet contains a grounded outer electrode and works with a pin-type powered electrode at a frequency of 1 MHz, inside a 1.6 mm-thin (inner diameter) dielectric ceramic tube. It has a dissipated power of 3.5 W, using a direct current power supply with a maximum system energy of 50 W at 100–240 V, 50/60 Hz.

### 2.4. In Vivo Experiments

For s.c. tumour cell injection, mice were anaesthetized (1.5–2.5% isoflurane (CP-pharma, Burgdorf, Germany) in oxygen) and kept on a heating pad (temperature of 38 °C). Both hind flanks were shaved, and 1 × 10^6^ A431 or A375 cells (in 1:1 cold DPBS/Matrigel^®^ HC GFR), respectively, were injected with s.c. in the left and right side.

Mice were divided into two groups of CAP- and non-CAP (control) treatment of 12 to 13 individuals, each per group and tumour cell line, respectively. In CAP-treated groups, potential pain and fixation stress under treatment was reduced with isoflurane anaesthesia (performed as mentioned above). CAP treatment started on day four after tumour cell injection and was repeated every four days until day 20. It was performed for 5 min in horizontal and vertical directions, line by line, in an area of at least 1 cm^2^ over the entire tumour. The plasma source was used at a 45° angle to reduce the risk of heat accumulation over the skin surface. To quantify local skin temperature during the treatment, an infrared thermometer (BodyTemp 478, Dostmann electronic GmbH, Reicholzheim, Germany) was used before and after CAP therapy. For analgesia, the animals received 0.25 mL metamizol (500 mg/mL, Ratiopharm GmbH, Ulm, Germany) in the drinking water (100 mL, water was changed daily) from the day of s.c. tumour cell injection, on until euthanasia. The tumour size was measured with a Calliper to study the tumour growth over time. Volume calculation was obtained using formula length × width^2^ × 0.52. For a more detailed assessment of tumour volume, longitudinal multimodal imaging techniques were applied. The use of such imaging techniques not only allows for close monitoring of parameters in individual animals but also leads to a significant reduction in the number of animals used, which is in direct agreement with the 3-R principle (reduce, replace, refine). After in vivo experiments, mice were anaesthetized by intraperitoneal (i.p.) injection of ketamine/xylazine (ketamine-hydrochloride (PHARMANAOVO GmbH, Hannover, Germany) 100 mg/kg; xylazin-hydrochloride (Bayer Vital GmbH, Leverkusen, Germany) 6 mg/kg) and blood was collected by a retroorbital puncture. Animals were subsequently sacrificed by cervical dislocation. Afterwards, tumours were harvested for further analysis. In the present study, 50 mice were used. In total *n* = 24 tumours were dissected from *N* = 12 mice in SCC, as well as in MM group and *n* = 26 tumours from *N* = 13 mice in SCC + CAP, as well as in MM + CAP group. Tumours that were not located s.c. were discarded from the study (*n* = 4 in SCC + CAP group and *n* = 2 in MM + CAP group). Mice were allocated in a non-random manner matching the age and body weight of CAP- and non-CAP-treated groups. It was not possible to blind the observers due to CAP treatment. The observers were blinded for all subsequent analysis and assessment of MRI, PET/CT, immunohistochemical and molecular biological samples and the resulting data. Last, the mice were allocated to the groups based on their registration number. All animal experiments were performed at least twice and time-independent to reduce confounding factors.

### 2.5. Scoring

The animal keepers cared for and monitored the health of the animals every day, and the severity of their condition was assessed on each experimental day throughout the whole observation period. In brief, scoring was subdivided into four categories (weight loss, appearance, consciousness, process-specific criteria) with a maximum score of 17. Animals with a single score of ≥5 or a total score of ≥7 were considered as severely burdened (humane endpoint). Further humane endpoints included weight loss of ≥20% and tumour volume of ≥2 cm^2^.

### 2.6. MRI

MR imaging was performed using a 7 Tesla small animal MRI scanner (BioSpec 70/30, 7.0 T, max. 440 mT/m gradient strength, Bruker BioSpin MRI GmbH, Ettlingen, Germany) with a 1 H transmit resonator (86 mm Resonator, Bruker, Billerica, MA, USA) and a 2-by-2 receive-only surface coil array (Bruker), positioned on the back of the mice. For MRI measurement, mice were anaesthetized with isoflurane (as mentioned above) and positioned on a heating pad. Respiration rate was monitored continuously (Model 1030 monitoring and gating system, SA Instruments, Stony Brook, NY, USA). MRI was performed one week after tumour cell injection and repeated every week for three weeks (three MRI scans per mouse in total). MR- and PET/CT imaging was performed on separate days to keep the anaesthesia time per day as short as possible. The imaging protocol included morphological T2-weighted TurboRARE and diffusion-weighted (DWI) SpinEcho imaging sequences. Tumour size was assessed in high-resolution T2-weighted images of the transversal plane. The T2-weighted TurboRARE sequence parameters were: repetition time (TR): due to respiratory gating approximately 4.200 ms; echo time (TE): 26.0 ms; field of view: 42 mm × 24 mm; matrix: 351 × 200; voxel size: (0.12 × 0.12 × 0.75) mm^3^; 35–50 slices depending on tumour size; acquisition time: approx. 10 min. SpinEcho DWI-sequence parameters were: TR: 2.000 ms; TE: 25 ms; field of view: 42 mm × 24 mm; matrix: 192 × 109; resolution: 220 × 220 µm, 35–50 slices of 0.75 mm per slice in the transversal plane; diffusion weighting four b values, b_1_–b_4_: (100, 350, 700, 1.000) s/mm^2^, one b_0_ image and three orthogonal gradient directions, total acquisition time: 30–35 min, depending on tumour dimensions and the number of slices. The apparent diffusion coefficients (ADC)-maps were calculated by a pixel-wise least-square monoexponential fit of the signal intensity for the different b-value images of the spin-echo DWI-sequence. Tumour volume and ADC values were analysed by slice wise region of interest (ROI) placement using ITK-SNAP software (v3.6.0; 1 April 2017, Penn Image Computing and Science Laboratory, University of Pennsylvania, Philadelphia, PA, USA).

### 2.7. PET/CT Imaging

PET/CT in vivo imaging of metabolic tumour volume (MTV) was performed with [^18^F]FDG. PET/CT imaging was performed equal to MRI on week one, two and three. For the tracer injection and imaging anaesthesia and maintenance of body temperature were performed as mentioned above. Prior to the scan, the tracer was injected into a tail vein via a microcatheter. Mice were measured in prone position in a small animal PET/CT scanner (Inveon PET/CT, Siemens Medical Solutions, Knoxville, TN, USA). For [^18^F]FDG scans, mice received a mean dose of 15.0 ± 2.0 MBq and after an uptake period of 60 min, mice were statically imaged for 15 min. For anatomical landmarking and attenuation correction, a whole-body CT scan for each mouse was performed. Correction of the PET data sets was carried out for random coincidences, scatter, time of death, decay, and attenuation. A Feldkamp algorithm was used to reconstruct CT images. Reconstruction of 3D PET images was performed with a 3-dimensional iterative ordered subset expectation maximization (3D-OSEM/OP-MAP) algorithm with 2 iterations (OSEM), 18 iterations (MAP) resulting in 1.5 mm spatial resolution. MTV was determined using Inveon Research Workplace (v4.2.08, Siemens, Knoxville, TN, USA). On fused PET and CT images, a volume of interest (VOI) was placed around the whole tumour tissue and the MTV was obtained by applying a threshold of 60% to the SUVmax to obtain the 40% of the hottest voxels in the VOI. Thus, partial volume effects, e.g., spill-in and spill-out effects, were reduced and necrotic parts of the tumour were excluded [22].

### 2.8. Luminescence Imaging

Luminescence imaging of ROS was performed with the chemical counterpart of luminol, the CL probe L-012 (8-amino-5-chloro-7-phenyl-pyrido [3,4-d] pyridazine-1,4(2H,3H)dione). L-012 (FUJIFILM Wako Pure Chemical Corporation, Neuss, Germany) was prepared in sterile aqua dest. and DPBS (250 mL/L) at a concentration of 10 mg/mL. Luminescence imaging of the mice was performed using a small animal imaging system (NightOWL II LB 983 NC100, Berthold Technologies GmbH & Co. KG, Bad Wildbad, Germany) with an enclosed, dark chamber and a CCD camera to capture L-012 signal. Prior to the imaging, L-012 was injected i.p. at a dose of 50 mg/kg. Luminescence imaging was performed in isoflurane-anaesthetized mice in prone position after an uptake period of 330 s. All groups were measured on day 4, 8, 12, 16 and 20. In addition, the CAP-treated group was imaged two hours prior to CAP treatment to allow for metabolic degradation L-012 in a second imaging session on the same day, immediately followed by CAP exposure and L-012 administration. For luminescence imaging, exposure times were 60 s with 4 × 4 binning. Background signal correction and cosmic suppression were performed for all images. To locate the tumours, a bright-field image was recorded with an exposure time of 0.01 s and light intensity of 10%. Luminescence signals were semi-quantitatively analysed by manually ROI placement over the area of the right tumour, using indiGO™ Software (v2.0.5.0, Berthold Technologies GmbH & Co. KG, Germany). Luminescence intensity from the ROI was quantified as photons/second/cm^2^.

### 2.9. Histology and Immunohistochemistry

Half of each tumour was fixed in Formafix, 4% buffered (stabilized with methanol; Grimm med. Logistik GmbH, Torgelow, Germany) for one day and prepared for paraffin embedding. For hematoxylin and eosin (H&E) staining, tumours were cut in a thickness of 4 µm. After deparaffining, staining was performed and tissue sections were assessed for routine histology. DNA fragments of apoptotic cells were enzyme-labelled and detected employing the ApopTag^®^ assay (Merck, Darmstadt, Germany), which is based on TUNEL method. Staining was performed according to the instructions for use, followed by counterstaining with Mayer’s hemalaun solution (Merck, Darmstadt, Germany). Ki-67 as marker for proliferating cells was labelled with a rabbit polyclonal anti-Ki-67 (1:200, Abcam, Cambridge, UK) and a secondary polyclonal goat anti-rabbit IgG with alkaline phosphatase (AP) conjugate (1:100, Agilent Dako, Santa Clara, CA, USA), followed by counterstaining with permanent red solution (Agilent Dako). Staining of endothelial CD31 was performed using a rat polyclonal anti-CD31 (PECAM-1, 1:50, Santa Cruz Biotechnology Inc., Dallas, TX, USA), a secondary polyclonal goat anti-rat IgG with horseradish peroxidase (HRP) conjugate (1:200, Abcam, Cambridge, UK) and the chromogen DAB (Agilent Dako), followed by counterstaining with Mayer’s hemalaun solution (Merck). Macrophages were detected with a rat monoclonal anti-F4/80 (1:10, Bio-Rad Laboratories Inc., Hercules, CA, USA) and secondary polyclonal goat anti-IgG with AP conjugate (1:100, Abcam). After specific labelling, slides were counterstained with permanent red solution (Agilent Dako). Chloracetate esterase (CAE) staining of deparaffined slides was used to detect granulocytes. Therefore, slides were stained with Naphthol AS-D chloroacetate (Sigma-Aldrich) mixed with diazonium salt (Sigma-Aldrich) and counterstained with Mayer’s hemalaun solution. All slides were embedded in X-TRA Kitt^®^ (MEDITE, Burgdorf, Germany). Ki-67, ApopTag^®^, F4/80 and CAE positive-stained cells were quantified using 400× magnification with the Axiskop 40 microscope (Zeiss, Jena, Germany) by evaluating 5 × 100 nuclei, located in five representative areas along the invasive vital front of the tumour, per tumour and given as positive cells [%]. The number of CD31-positive vessels were counted in five fields of view and given as number of vessels/field.

### 2.10. Molecular Biology

For the analysis of protein expression, half of the tumours were snap-frozen in liquid N_2_ (−196 °C) immediately after resection and stored at −80 °C. 30 mg per tumour were manually homogenized, and proteins were isolated with lysis buffer (1 M Tris pH 7.5, 5 M NaCl, 250 mM EDTA, 10% Triton-X-100, 4% NaN_3_, 100 mM PMSF (a protease inhibitor cocktail)). After separation of 20 µg or 60 µg (for cleaved-caspase-3) protein by SDS-PAGE on Tris-glycine gels, proteins were transferred to polyvinyl difluoride membranes (Bio-Rad Laboratories). Membranes were blocked with 1× TBS-T plus 2.5% BSA at 4 °C overnight with following antibodies: a mouse monoclonal anti-PCNA (1:5000, Abcam, Cambridge, UK), a rabbit polyclonal anti-caspase-3 (1:1500, Stressgen Biotechnologies, San Diego, CA, USA), a rabbit polyclonal anti-cleaved-caspase-3 (1:1000, BD Heidelberg, Deutschland) and a mouse monoclonal anti-β-actin (1:20,000; Sigma-Aldrich). Subsequently, secondary peroxidase-linked anti-mouse antibodies (PCNA; 1:10,000; β-actin; 1:60,000) or anti-rabbit antibodies (caspase-3, 1:8000; cleaved-caspase-3, 1:2000) were used. Protein detection was performed by luminol-enhanced CL (ECL plus; Thermo Fisher Scientific). Imaging of protein expression was detected with ChemiDoc™ XRS+ System (Bio-Rad Laboratories). Luminescence signals were assessed, normalized to β-actin, and relatively quantified with Image Lab™ software (v6.0.0 build 25, Bio-Rad Laboratories).

### 2.11. Statistical Analysis

Values were expressed as mean ± standard deviation (SD) for the indicated number of samples. Data visualization and statistics were performed using GraphPad Prism Software (v8.4.3 (686), GraphPad Software, La Jolla, CA, USA). Two-way repeated measurement ANOVA with Geisser-Greenhouse correction was performed to evaluate changes within a group over time. Comparisons of treated or untreated groups were conducted employing two-way ANOVA or the mixed-effects analysis model for groups with missing values, including adjustments of *p*-values, using Bonferroni correction. Comparisons of multiple time-independent groups were performed using the one-way ANOVA, followed by Tukey’s multiple comparison. To compare the two groups, the unpaired two-tailed *t*-test was used. *p* values of ≤0.05 were considered statistically significant. Significance levels are abbreviated as follows: * *p* ≤ 0.05, ** *p* ≤ 0.01, *** *p* ≤ 0.001, **** *p* ≤ 0.0001.

## 3. Results

### 3.1. Characterization of SCC and MM

After s.c. tumour cell injection of either SCC or MM in the flanks of NSG mice, both entities formed macroscopically visible tumours within the following days (take rate = 100%). All mice with MM survived the time course of the study, while two mice of the untreated SCC group and one mouse of the CAP treated SCC group died. Upon detailed inspection, systemic metastases could be excluded as a reason for death.

Over the three-week observation, tumours were detectable by MR and PET/CT imaging (Figure 1A,B). Both imaging techniques showed a rapid increase in the tumour volume and an inhomogeneous tumour structure. SCC especially depicted a rather inhomogeneous structure with a massive central necrosis area and viable tumour cells in the periphery. These observations were verified with H&E staining of tumour tissue at the end of the in vivo experiments (Figure 1C,D). In line with this, Ki-67-positive cells were mainly found in the periphery of tumours, as well as thin vessels with a small diameter formed by CD31-positive cells, whereas apoptotic cells were found in the border zone between central necrosis and periphery (Figure 1C,D).

### 3.2. Tumour Volume under CAP Treatment

CAP treatment was started on day four after s.c. tumour implantation and repeated every four days. To assess the effect of CAP therapy on tumour growth, tumour volumes were measured by both Calliper and MRI. Calliper data are given in Appendix A (Appendix A). T2-weighted MRI revealed significantly increased volumes of both tumour entities from the treated and untreated group over the observation time (*p* ≤ 0.05, respectively). CAP therapy caused a significant reduction in anatomical tumour volume in week one, two and three in both tumour entities, compared to none treated tumours. Tumour growth, both in MM and in SCC, was decreased by approximately 50% at week three, due to CAP treatment (Figure 2A–D).

In addition, diffusion-weighted MR imaging was performed to assess the cellular density within the tumour. Both tumour entities showed slightly decreasing ADC values over time (Table 1) reflecting the evolution to solid tumours. Interestingly, at all three time points (week one, two and three) CAP-treated tumours showed higher ADC values when compared to untreated tumours (Table 1). This data implies a reduction in cellular density with an increase in free water, and points towards the therapeutic efficacy of CAP.

In order to assess the effect of CAP on the MTV of the two tumour entities, we carried out PET/CT examinations using [^18^F]FDG radiotracer (Figure 3A–D). Three weeks after CAP treatment, no effect on MTV could be determined in SCC (Figure 3B). In contrast, CAP treatment reduced the MTV in MM over time (Figure 3D).

### 3.3. CAP Treatment Increased Reactive Species In Vivo

To evaluate reactive species-mediated antitumour effects of CAP therapy, the CL probe L-012 was used. The luminescence intensity was measured in untreated mice, as well as in CAP-treated ones. In the latter, CL intensity was assessed both before and after CAP treatment (Figure 4A–D). In both tumour entities, CAP treatment significantly increased reactive species within the tumour. In contrast, untreated tumours and tumours before CAP treatment were characterized by negligible CL signal upon administration of L-012 (Figure 4B,D).

### 3.4. Effect of CAP Treatment on Skin Temperature

To assess whether CAP treatment affects the tumour growth by local heating, skin surface temperature at the tumours was measured before and immediately after treatment. During CAP treatment, the mean skin temperature increased by 4.5 ± 2.0 °C compared to normal skin temperature of 32.2 ± 2.5 °C and reached a maximum of 37.8 ± 2.8 °C. Hence, mild hyperthermia might have contributed to the anticancer effects of CAP.

### 3.5. Molecular and Histological Tumour Analysis

Immunohistochemical evaluation revealed a significant reduction in Ki-67-positive cells after CAP treatment of SCC (*p* < 0.01 vs. no CAP treatment) (Figure 5A). In contrast, no differences with respect to the presence of apoptotic cells, CD31-positive vessels, CAE-positive granulocytes, and F4/80-positive macrophages between untreated and CAP-treated tumours were found (Figure 5A,B). In line with these findings, analysis of PCNA, caspase-3 and cleaved-caspase-3 protein expression depicted no significant changes between treated or untreated tumours (Figure 5C–F; Appendix A).

## 4. Discussion

This study demonstrated that CAP treatment reduces the volume increase in cutaneous skin tumours over the course of three weeks (i). In parallel, tumour cellularity was reduced in MM (ii). Based on the performed analysis, it can be concluded that these effects are mainly mediated through the formation of ROS during CAP therapy (iii). Despite a reduced proliferation in SCC, CAP neither affected apoptosis (iv), inflammation (v), angiogenesis (vi), nor metabolic activity (vii) in MM and SCC, respectively.

This was the first time that multimodal imaging techniques, such as MRI and PET/CT, were used to assess the tumour progression in vivo in a longitudinal and non-invasive manner, and to monitor the therapeutic efficiency of CAP. Multimodal imaging techniques are ideally suited for evaluating antitumour therapeutic efficacy in animal models because of its non-invasive nature [23]. Complementing benefits include longitudinal in vivo imaging of growth kinetics, metabolism, metastasis, and cellularity of tumours. Hence, these imaging methods play a critical role in reducing the number of animals needed and their burden, which is in line with the 3-R principle of international animal welfare recommendation.

Although CAP treatment did not lead to a tumour remission, the growth of MM and SCC was significantly reduced after one, two and three weeks, respectively. It is reasonable to assume that this growth inhibition is related to ROS and RNS formation in tumour tissue after CAP treatment, which was confirmed by a significant increase in the luminescence signal of L-012. To our knowledge, this study proved for the first time that the CL probe L-012 is ideal for non-invasive detection of reactive species in tumour tissue during CAP treatment in living mice.

Reactive species induce the oxidation of L-012, which subsequently leads to an increase in CL [24]. Compared to other probes, L-012 shows a higher sensitivity to ROS and RNS, which has already been confirmed by several studies [25,26] and was also used for different in vivo experiments [27,28]. By detecting ROS, the present study confirmed that ROS resulting from CAP treatment overcomes tissue barriers and can be used for tumour control. This is in line with findings showing that ROS and RNS could penetrate a 1 mm thick gelatine film in vitro [29], while in vivo it was shown that CAP-derived ROS and RNS oxidize and pass the stratum corneum to subsequently elicit intracutaneous responses [30,31]. To confirm that ROS and RNS mediate the anticancer effect of CAP, it was shown that treatment of MM with CAP in the presence of glycerol and sodium bicarbonate gel, used as scavenger for reactive species, radicals, and ions, inhibits the ability of plasma to ablate the tumour, which vice versa leads to tumour reoccurrence [32]. ROS are assumed to result in tumour cell apoptosis, which could not be confirmed in the current study. This observation also contrasts other studies, showing an increased number of apoptotic cells in patient-derived human melanoma tissue [33] and cell cycle arrest after CAP treatment [34]. Moreover, CAP treatment reduced cell viability in murine mesenchymal stem cells. Here, cell lysis by reactive species and charged particles was considered one of the primary mechanisms causing cell death [35]. It is further suggested that treatment with CAP affects the cell cycle of tumour cells to a greater extent than non-tumour cells, due to increased reporters for oxidative stress in the S phase [36]. One explanation for this apparent discrepancy might be the penetration depth of ROS. It was reported that reactive species generated by CAP can only overcome a short distance from tissue barriers [37]. Thus, the s.c. heterotopic tumour model in this study and not intracutaneous orthotopic location might explain that typical ROS-expected effects, such as increased tumour cell apoptosis, could not be detected. Dedicated analysis of tumour areas directly affected by CAP treatment could possibly provide better information than the consideration of CAP effects on the entire tumour.

Nevertheless, tumour cells might be affected upon the cumulative effect of ROS/RNS, UV radiation, as well as electric fields, and undergo a variety of cell death forms, such as necrosis or regulated cell death forms, such as intrinsic or extrinsic apoptosis, necroptosis, pyroptosis, and others [38], which were not addressed explicitly by our methodological spectrum. This assumption is supported by the constantly higher, though not significant, ADC values in CAP-treated tumours. The therapeutic effectiveness of CAP treatment in MM was substantiated by the reduction in cellularity in week three, i.e., a significant increase in ADC values, while ADC values only slightly increased in CAP-treated SCC mice. This is in line with data from Eschbach et al. who also showed that tumour therapy with BRAF and CDK 4/6 inhibitors increased the ADC values of MM in mice, compared to non-treated tumours [39]. Furthermore, reduced cellularity in tumours, i.e., an increase in ADC values, is an established parameter for the assessment of a therapeutic response in humans [40].

In this study, MTV determined by ^18^[F]FDG PET/CT showed no significant decrease under CAP treatment, although a tendency could be observed in MM. The reduced MTV could be an indication of the therapeutic effect of the CAP treatment, similar to the increase in ADC values after the CAP treatment. It has already been shown in an in vivo study with MM that the treatment response was indicated by an increased ADC value and also a reduced uptake of ^18^[F]FDG [39]. It cannot be excluded that partial volume effects, or the limited spatial resolution of the PET could have obscured a decrease in MTV after CAP treatment, although we only included the 40% of the hottest voxels in the tumour VOI to reduce partial volume effects. Furthermore, a biased selection of the VOI, due to manual placement, could have occurred. However, fused PET and CT images and visual inspection in all three planes were used for VOI placement to reduce any observer bias.

Since hyperthermia revealed a direct anticancer effect at local temperature between 41 °C and 47 °C in vitro and in vivo [41], the CAP-induced heating of the skin was assessed as well. The slight temperature increase upon CAP treatment in this study might have contributed to the described tumour growth inhibition.

When analysing the results of this study, the mouse model also has to be considered an individual factor. NSG mice do not mount immune responses against tumours due to IL-2R common gamma-chain mutation, which results in a deficiency of NK cells and mature lymphocytes [42,43]. Therefore, it is impossible to assess the potential influences of the immune system on tumour development in this model. However, the use of these mice was inevitable to allow human tumour cell lines to grow, which generally allows to translate the findings of this study into humane biology. A recent clinical study focused on the effects of myeloid cells and immune-mediated cell death in relation to the mechanistic effect of CAP therapy on tumours. It was found that the number of CD11b^+^ myeloid cells in a biopsy from patients with plasma-treated tumour tissue is decreased compared to untreated patients, which was associated with a better outcome [44]. This led to the hypothesis of an immunological dimension of CAP treatment in oncology that is driven by myeloid antigen-presenting cells to promote antitumour adaptive immunity [45,46]. Along those lines, we have previously shown increased myeloid and lymphoid immuno-infiltration into gas plasma-treated syngeneic B16 melanomas in vivo [47], while checkpoint immunotherapy augmented therapeutic efficacy in the similar tumour model, but using another plasma device [48]. An in vitro study using the same two tumour cell lines showed that not only the adaptive immune system mediates anti-cancer effects after CAP treatment, but also innate immune cells, such as NK cells, lead to significantly higher apoptosis of tumour cells, due to expression modulation of activating and inhibiting receptors on tumour cells after CAP treatment [49].

In summary, it can be assumed that CAP is a promising treatment option for cancer therapy and offers the possibility of treating topically localized tumours. The obtained data also show that the formation of reactive species in tumours upon CAP treatment can be directly detected in vivo, using the L-012 CL probe. Due to reduced growth kinetics of both tumour entities, it can be assumed that the effect of CAP is mediated locally so that this technology might be used as a treatment option for superficial tumours, or more likely as adjuvant therapy in addition to established standard therapies, such as local resection of skin cancer. After resection, CAP could also be used to treat the resection margin to improve a complete local removal of malignant cells. However, to add CAP as a treatment option to existing cancer therapies, further pre-clinical and clinical trials need to be conducted. A first step could be to use orthotropic mouse models with an intact immune system and local skin tumours to better investigate and understand the influence of treatment on the immune system-mediated anticancer effects. One possibility would be the humanization of NSG mice with lymphoid and myeloid cells.

## Figures and Tables

**Figure 1 cancers-13-02483-f001:**
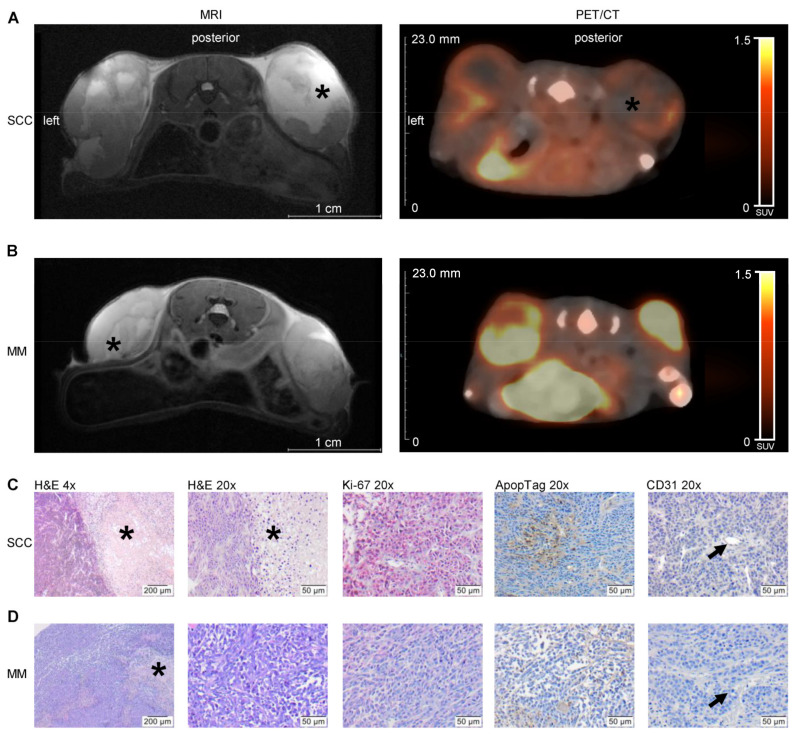
Representative MR (T2 weighted, axial, week three) and PET/CT images ([^18^F]FDG; summed images, axial slices, week three) (**A**; SCC and **B**; MM) as well as histological and immunohistochemical images of H&E, Ki-67, ApopTag^®^ and CD31 staining (**C**; SCC and **D**; MM) for characterization of tumour biology. In NSG mice 1 × 10^6^ tumour cells of either squamous cell carcinoma (SCC) or malignant melanoma (MM) were injected s.c. into the left and right hind flank. Over the three-weeks observation time, anatomical tumour volumes were measured by MRI, and metabolic tumour volumes were determined by PET/CT. Measurements were performed one, two and three weeks after tumour cell implantation. After in vivo characterization, the tumours were excised and prepared for histological analysis. Asterisks mark central tumour necrosis. Arrows point to CD31-positive vessels.

**Figure 2 cancers-13-02483-f002:**
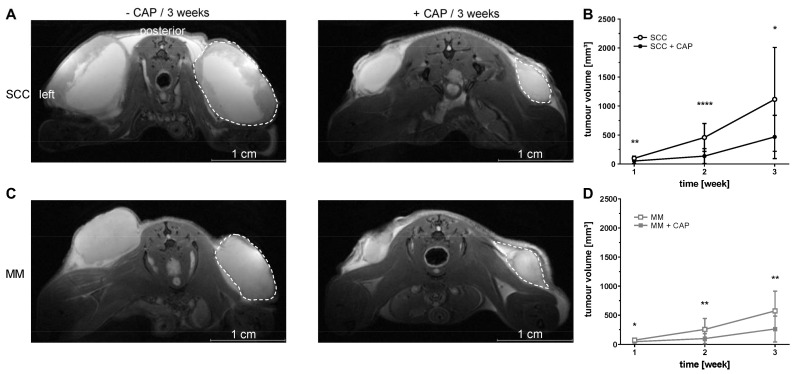
Representative MR images (**A**,**C**; T2 weighted, axial) and quantitative assessment of tumour volumes (**B**,**D**). NSG mice with s.c. flank tumours, either squamous cell carcinoma (SCC) or malignant melanoma (MM), were treated with or without cold atmospheric pressure plasma (CAP) over three weeks. CAP treatment started four days after tumour cell injection and was repeated every four days. CAP treatment caused a significant reduction in both SCC and MM growth over the three-week observation time. Empty symbols represent untreated tumours, and filled symbols represent CAP-treated tumours. Data are presented as mean ± SD (*n* = 20–24 samples per time point); two-way ANOVA followed by Bonferroni correction for multiple comparison. * *p* ≤ 0.05; ** *p* ≤ 0.01; **** *p* ≤ 0.0001 vs. untreated SCC or MM, respectively. Dotted line mark right tumour volume.

**Figure 3 cancers-13-02483-f003:**
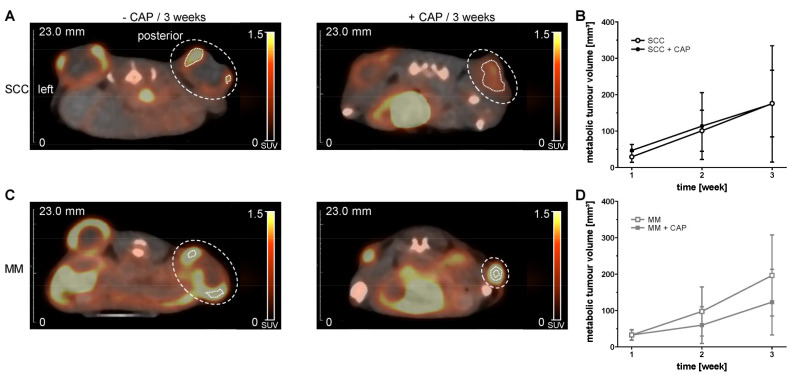
Representative PET/CT images from [^18^F]FDG (**A**,**C**; summed images, axial slices) and quantitative assessment of metabolic tumour volumes (**B**,**D**). NSG mice with two s.c. flank tumours, either squamous cell carcinoma (SCC) or malignant melanoma (MM), were treated with or without cold atmospheric pressure plasma (CAP) over three weeks. CAP treatment started four days after tumour cell injection and was repeated every four days. Data are presented as mean ± SD (*n* = 6–20 samples per time point); mixed-effects analysis, followed by Bonferroni correction. Large, dotted line mark manually placed volume of interest. Short, dotted line mark right metabolic tumour volume.

**Figure 4 cancers-13-02483-f004:**
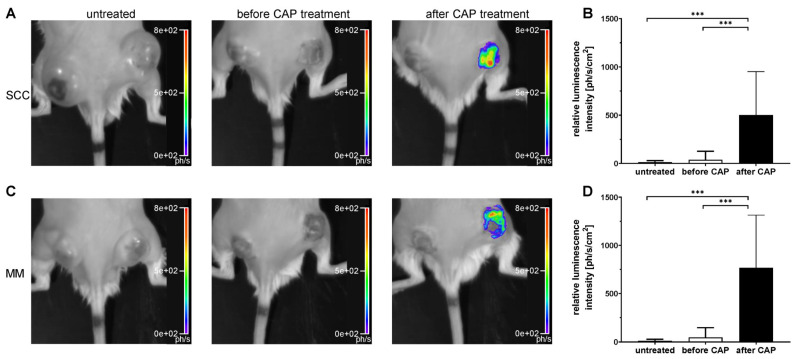
Representative luminescence images (**A**,**C**; luminescence probe: L-012) and quantitative assessment of relative luminescence intensity (**B**,**D**). NSG mice with two s.c. flank tumours, either squamous cell carcinoma (SCC) or malignant melanoma (MM), were treated with or without cold atmospheric pressure plasma (CAP) over three weeks. CAP treatment started four days after tumour cell injection and was repeated every four days. L-012 was injected i.p. and the luminescence intensity was measured before and after CAP treatment. The measurement was performed two hours before and immediately after the CAP treatment. Data are presented as mean ± SD (*n* = 10–15); one-way ANOVA followed by Tukey’s multiple comparison. *** *p* ≤ 0.001 vs. untreated SCC or MM, respectively or before CAP treatment of SCC or MM, respectively.

**Figure 5 cancers-13-02483-f005:**
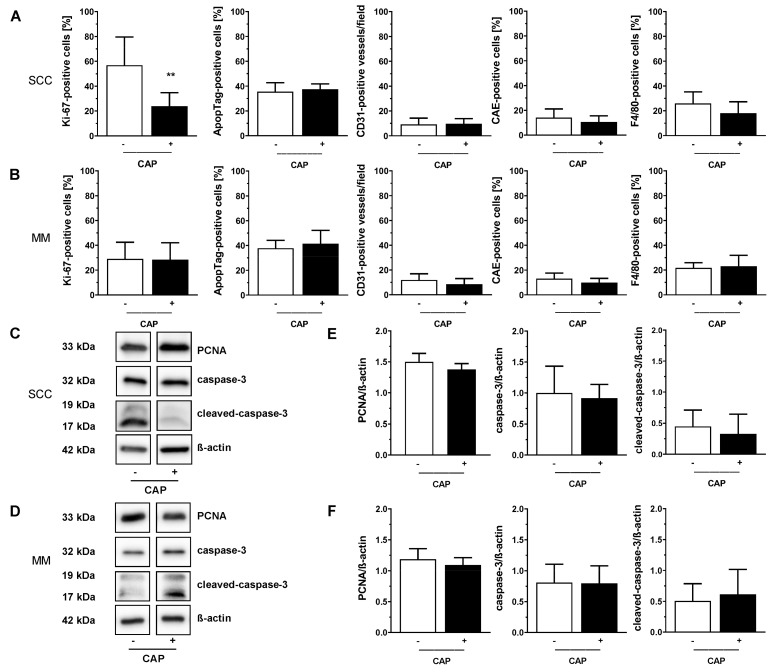
Quantitative immunohistochemical assessment of Ki-67, ApopTag^®^, CD31, CAE and F4/80 expression (**A**,**B**) as well as representative images (**C**,**D**) and quantitative western blot analysis of PCNA, caspase-3 and cleaved-caspase-3 (**E**,**F**) in SCC and MM, respectively. NSG mice with two s.c. flank tumours, either squamous cell carcinoma (SCC) or malignant melanoma (MM), were treated with or without cold atmospheric pressure plasma (CAP) over three weeks. CAP treatment started four days after tumour cell injection and was repeated every four days. After three weeks, tumours were excised and prepared for immunohistochemical and molecular biological analysis. Data are presented as mean ± SD (immunohistochemistry: *n* = 8–14; molecular biology: *n* = 5–8); unpaired two-tailed *t*-test. ** *p* ≤ 0.01 vs. untreated SCC.

**Table 1 cancers-13-02483-t001:** Data of apparent diffusion coefficient, assessed from diffusion-weighted MR imaging. NSG mice with two s.c. flank tumours, either squamous cell carcinoma (SCC) or malignant melanoma (MM), were treated with or without cold atmospheric pressure plasma (CAP) over three weeks. CAP treatment started four days after tumour cell injection and was repeated every four days. Data are presented as mean ± SD (*n* = 12–20 samples per time point); Mixed-effects analysis followed by Bonferroni correction for multiple comparison.

Apparent Diffusion Coefficient [×10^−3^ mm^2^/s]					
		−CAP (Mean ± SD)		+CAP (Mean ± SD)	
SCC	week 1	1.13 ± 0.17		1.16 ± 0.16	
	week 2	1.01 ± 0.18	to week 1 ##	1.07 ± 0.10	
	week 3	1.07 ± 0.26		1.19 ± 0.18	
MM	week 1	1.35 ± 0.23		1.44 ± 0.30	
	week 2	1.02 ± 0.19	to week 1 ####	1.15 ± 0.19	to week 1 #
	week 3	0.84 ± 0.09	to week 1 ####; to week 2 ###	1.03 ± 0.13	to week 1 ###; to week 2 ##; ****

**** *p* ≤ 0.0001 vs. untreated MM, respectively. ^#^
*p* ≤ 0.05; ^##^
*p* ≤ 0.01; ^###^
*p* ≤ 0.001; ^####^
*p* ≤ 0.0001.

## Data Availability

The datasets generated and analysed during the current study are available from the corresponding author on reasonable request.

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
