# Peer review of "Multimodal Imaging Techniques to Evaluate the Anticancer Effect of Cold Atmospheric Pressure Plasma"

_cancers, 2021, doi:10.3390/cancers13102483_

Round 1
Reviewer 1 Report
- Line 37 "Estimates assume that there will be over 1 milion new cases of NMSC in 2018 alone..." Year 2018 should be replaced by the year that will come or present year?? Sorry, I do not understand this sentence.
- Please provide the current voltage characteristics of CAPP in section 2.3.
- How was the plasma treatment time estimated for all in vivo experiments? Were these 5 minutes long treatments optimized in any previous article or based on any preliminary data?
- Line 426 should be "growth inhibition" instead of "groth"
- In my opinion the authors should include in the discussion paragraph one more paper published by Ramona CLEMEN et al. entitled Physical Plasma-Treated Skin Cancer Cells Amplify Tumor Cytotoxicity of Human Natural Killer (NK) Cells (DOI: 10.3390/cancers12123575). Eventhough, it is self-citation paper, it would explain the reason why there was no effect on tumor cells apoptosis as well as on tumor metabolic activity observed in performed in vivo experiments. Based on the information included in that paper it seemed that both human cancer cell lines used in reviewed paper become more sensitive for NK cells activity after cold plasma treatment. Since NSG mice do not induce an immune response against tumor (lack of NK cells and other lymphocytes) this biological effect could not be estimated in NSG mice bearing tumor of human skin cancer cell lines A375 and A431 origin. Therefore, it would be very difficult to verify the anti-tumor activity pathway induced after CAPP treatment.
- Reference should be prepared as it is mentioned in the Reference List and Citations Guide. Please note, that in the text, reference numbers should be placed in square brackets [ ], and placed before the punctuation; for example [1], [1–3] or [1,3].
Reviewer 2 Report
This paper reports about the potential use of plasmas for cancer treatment. The authors
assessed the effects of cold atmospheric pressure plasma (CAP) on SCC and MM in vivo, employing a multimodal imaging techniques. Longitudinal MR and PET/CT imaging were performed to determine the anatomic and metabolic tumour volume over three-weeks in vivo. The findings propose CAP as a potential adjuvant therapy option to established standard therapies of skin cancer.
It is an interesting paper, contains novel aspects which provides interesting results with appropriate references, however I have some questions and comments below:
- The figures should be improved regarding is quality, in particular, the axis numbers/text, which is too small.
- There are some typo errors, for example in line 106, “exept” it should be replaced by “except”
- In all manuscript, where it is written milliliters, it should appear mL, instead of ml.
- Also in the materials methods section I would like to see details regarding the power supply that was used and what were the conditions.
- Why the authors decided to use 5 min for CAP treatment? Did the authors try other exposure times?
- Also should be interesting to see the effect of the CAP treatment in normal cells in vivo. Did the authors perform some tests regarding normal cells?
